# Effect of Hollow 304 Stainless Steel Fiber on Corrosion Resistance and Mechanical Properties of Ultra-High Performance Concrete (UHPC)

**DOI:** 10.3390/ma16103612

**Published:** 2023-05-09

**Authors:** Tianran Li, Yulong Yan, Chengying Xu, Xiangnan Han, Yang Liu, Haiquan Qi, Yang Ming

**Affiliations:** 1School of Materials Science and Engineering, Guilin University of Technology, Guilin 541000, China; 2School of Civil and Architectural Engineering, Guilin University of Technology, Guilin 541000, China

**Keywords:** UHPC, hollow steel fiber, mechanical property, durability, electrochemical test, fiber network structure

## Abstract

This study investigated the effect of hollow 304 stainless-steel fiber on the corrosion resistance and mechanical properties of ultra-high-performance concrete (UHPC), and prepared copper-coated-fiber-reinforced UHPC as the control group. The electrochemical performance of the prepared UHPC was compared with the results of X-ray computed tomography (X-CT). The results reveal that cavitation can improve the distribution of steel fibers in the UHPC. Compared with solid steel fibers, the compressive strength of UHPC with hollow stainless-steel fibers did not exhibit significant change, but the maximum flexural strength increased by 45.2% (2 vol% content, length–diameter ratio of 60). Hollow stainless-steel fiber could better improve the durability of UHPC compared with copper-plated steel fiber, and the gap between the two continued to increase as the durability test progressed. After the dry–wet cycle test, the flexural strength of the copper-coated-fiber-reinforced UHPC was 26 MPa, marking a decrease of 21.9%, while the flexural strength of the UHPC mixed with hollow stainless-steel fibers was 40.1 MPa, marking a decrease of only 5.6%. When the salt spray test had run for seven days, the difference in the flexural strength between the two was 18.4%, but when the test ended (180 days), the difference increased to 34%. The electrochemical performance of the hollow stainless-steel fiber improved, owing to the small carrying capacity of the hollow structure, and more uniform distribution in the UHPC and lower interconnection probability were achieved. According to the AC impedance test results, the charge transfer impedance of the UHPC doped with solid steel fiber is 5.8 KΩ, while that of the UHPC doped with hollow stainless-steel fiber is 8.8 KΩ.

## 1. Introduction

With the development of society, the requirements for concrete in the field of construction engineering are becoming increasingly higher, and ultra-high performance concrete (UHPC) has emerged to satisfy these needs. In UHPC, the mixing of concrete particles is optimized by adding silica fume (nano-silica) and other fillers instead of coarse aggregate. In concrete, silica works as a nucleation site, accelerating the cement hydration and filling the voids, which results in higher packing density and lower porosity [1]. Moreover, silica can also react with calcium hydroxide [2] to form calcium silicate. Therefore, owing to its advantages of high strength and good durability [3,4], UHPC has been widely used in bridges, high-rise buildings, and other projects in recent years.

Although UHPC has good mechanical strength and durability, to reduce the porosity, the water binder ratio of UHPC is very low, which essentially makes UHPC a brittle material [5]. To achieve higher toughness and improve the practicability of UHPC, fiber is typically added to UHPC [6,7]. This type of UHPC is called ultra-high performance fiber-reinforced concrete (UHPFRC). The addition of fiber not only improves the toughness of UHPFRC, but also improves its mechanical strength. After the addition of fiber, UHPFRC exhibits strain hardening behavior [8]. The strength of concrete is not only expressed by the strength of the matrix. After the initial fracture of the concrete matrix, the introduced fiber plays a bridging role to prevent the continuous expansion of cracks. Moreover, the addition of fiber improves the compactness of concrete, and makes it more difficult for external corrosive substances to enter the concrete, which further improves the durability of UHPC [9].

The types of fiber added to UHPC include glass fiber, steel fiber, basalt fiber, polymer fiber, and so on [10,11,12,13,14,15,16,17,18]. Jiang et al. [10] found that with the increase of copper-plated steel fiber content from 1% to 6%, the porosity of UHPC gradually decreased from 18.4% to 8.3%. Ren et al. [11] demonstrated that the failure mode of a sample with sisal fiber changed from brittle failure to toughness. Compared with an ordinary UHPC sample, when 2% sisal fiber (length of 18 mm) was added, the bending strength and toughness of the UHPC increased by 16.7% and 540.0%, respectively. The addition of polymer fiber, such as polypropylene fiber, can greatly improve the toughness of UHPC, but the improvement on the mechanical strength of UHPC is not ideal. Lai et al. [12] found that the simultaneous use of steel fiber and basalt fiber produced a synergistic effect, which greatly improved the impact and explosion resistance of the sample. Bei et al. [19] found that after cryogenic attack, the flexural strength of UHPC increased by 70.06% with the increase of length-to-diameter ratio of steel fiber, and the peak deflection increased from 0.501 mm, 0.919 mm to 0.609 mm, 1.302 mm.

However, the addition of steel fiber greatly increases the cost of UHPC. Therefore, many studies have attempted to reduce the amount of steel fiber without reducing the performance of UHPC so as to control the cost. Hui et al. [20] used 0.5% recycled tire steel fibers instead of industrial steel fibers, and achieved excellent synergistic effects, resulting in the highest dynamic splitting tensile property of UHPC at a strain rate of approximately 4.5–6.5 s^−1^, while reducing the material cost, carbon content, and energy content of UHPC by 9–57%. From practical and economic viewpoints, steel fiber is currently the most used fiber, and can greatly improve the mechanical properties of UHPC compared with other fibers [21,22]. The mechanical properties of UHPFRC are mainly affected by the amount of steel fiber [23], length–diameter ratio [6], fiber shape, and fiber distribution and orientation. According to a study by Wu et al. [21], compared with UHPC without any fiber, 1–3% straight steel fiber content increased the compressive strength and flexural strength by 8–32% and 22–72%, respectively. Moreover, compared with a reference mixture containing the same amount of straight fiber, the flexural strength of the reference mixture increased by 4–10% and 10–27%, respectively, after adding 1–3% corrugated fiber and hook fiber. Gesoglu et al. [22] found that the fracture energy of concrete increased by 101 times, 601 times, and 50 times when using micro-steel-fiber-, hook-steel-fiber-, and micro-glass-fiber-reinforced concrete, respectively. Yoo et al. [24] found that the flexural strength of fiber with an aspect ratio of 97.5 was higher by 40% compared with that of fiber with an aspect ratio of 65 at 2% by volume. However, the flexural strength of the fiber with the aspect ratio of 100 was lower by 20% compared with that of the fiber with the aspect ratio of 97.5. The main reason for this is that, when many long fibers are included, the interaction between the fibers interferes with the fiber arrangement in the tensile load direction. By investigating the influence of the concrete pouring height, pouring length, and viscosity, Song et al. [25] found that the concrete pouring height had the greatest influence on the fiber distribution and orientation in UHPFRC.

Steel fiber mainly affects the mechanical properties of UHPFRC through the bond strength with the concrete matrix [26]. The mechanism by which the fiber addition can effectively improve the toughness of the specimen is as follows: after micro cracks appear in the UHPFRC specimen, the released deformation energy is first used for fiber debonding instead of supporting the crack’s continued expansion, which delays the fracture process and achieves a toughening effect [27]. Therefore, the strengthening effect of fiber on the mechanical properties of UHPFRC can be quantified as the bonding performance of the fiber and concrete matrix. For example, long fiber is more effective compared with short fiber, because the bonding area of long fiber and the concrete matrix is larger [28,29]. Owing to the shape change of shaped fiber types (end hook type, wave type, and so on), the anchoring effect of shaped fiber is stronger than that of straight fiber and the concrete matrix; therefore, the improvement of the mechanical properties of UHPFRC is more obvious [30]. In addition to the above-mentioned methods for changing the bonding strength, various scholars are currently investigating the improvement of bonding performance with a concrete matrix by changing the surface state of steel fiber. Relevant methods include the chemical solution immersion method [31,32], sandpaper polishing method [33,34], and nano silica coating. The chemical solution immersion method modifies steel fiber by placing it in solutions such as ethylenediaminetetraacetic acid (EDTA) to increase the fiber surface roughness with the objective of improving the tensile strength of UHPC. The surface roughness parameter can increase 10 times and the tensile strength can increase by 36% [31]. The sandpaper polishing method uses sandpaper with different particle sizes to polish the fiber and increase its surface roughness with the objective of increasing the bonding performance. After sanding, the fibers are typically combined with PVA fibers to improve the mechanical properties [34]. Pi et al. [35] coated the steel fiber surface with nano-SiO_2_ multilayer film, and the SiO_2_ on the fiber surface reacted with Ca(OH)_2_ in the concrete. Thus, the interface transition zone between the fiber and the matrix formed a denser microstructure and the UHPC performance improved. From the above-mentioned studies, it is understood that the improvement of the mechanical properties of UHPC by physical modification is not as good as that by chemical modification.

Owing to the compactness of UHPC, the durability of steel fiber concrete can also be guaranteed. However, the main corrosion area of UHPFRC is the interface transition zone between the steel fiber and the concrete matrix. When the steel fiber content reaches a certain level, the durability of UHPFRC may decline, because an excessive amount of steel fiber leads to the formation of a fiber network in the concrete matrix, resulting in the connection of steel fibers. Therefore, the interface transition zone is also combined and connected through the concrete, and thus provides channels for corrosive ions, such as chloride ions, and results in electrochemical corrosion [36]. To prevent the above-mentioned situation, it is necessary to investigate the electrochemical performance of UHPC.

The objective of this study was to reduce the cost of UHPC by using steel fibers with cavitation structures instead of using solid steel fibers to reduce the use of materials. To improve the durability of UHPC in corrosive environments, such as the ocean, 304 stainless steel was used as the steel fiber material. Existing studies have mainly focused on the impact of steel fibers in solid structures on the performance of UHPC, while research on steel fibers in hollow structures is lacking. There is also limited research on stainless-steel fibers. This study investigated the effect of hollow stainless-steel fiber on the corrosion resistance and mechanical properties of UHPC. First, the effect of hollow stainless-steel fiber on the mechanical properties of UHPC was investigated by comparison to ordinary steel fiber. The influence of hollow stainless-steel fiber on the corrosion resistance of UHPC was investigated by dry–wet cycling and salt spray corrosion testing. Moreover, the electrochemical properties of UHPC with hollow stainless-steel fiber and the hollow stainless-steel fiber distribution were tested by X-ray computed tomography (X-CT).

## 2. Materials and Methods

### 2.1. Materials

The cementitious materials used in this study were Portland cement (P·O425) and silica fume. Three types of quartz sand (20-grain, 40-grain, and 80-grain) with different particle sizes were used as aggregates. Additionally, a water reducer and defoamer were added to improve the performance of UHPC (see Table 1 for the amount of UHPC materials). Three steel fiber types were used: hollow stainless-steel fiber, solid stainless-steel fiber, and copper-plated steel fiber, as illustrated in Figure 1. The properties of the three types of steel fiber are summarized in Table 2.

### 2.2. Experimental Method

#### 2.2.1. Mechanical Property Test

After all materials were mixed evenly, the mix was poured into a 40 × 40 × 160 mm mold to cure for 1 day. The specific preparation process is shown in Figure 2. The mold was removed, and the mix was then placed in a curing room (temperature of 20 °C and humidity of 95%) to cure up to the specified age.

The flexural strength of the sample was tested using a three-point bending test with a span of 100 mm and a loading rate of 50 N/s. In the compressive strength test of the UHPC, the loading rate was 2400 N/s, as shown in Figure 3. Three samples were used in each test. The flexural strength and compressive strength of the UHPC were measured according to the Chinese national standard GB/T 39147-2020. Data were obtained by averaging the test results of the three samples. When the data fluctuated over 15%, the data were rounded off. Moreover, when there were two data fluctuations exceeding 15%, the group of experiments was repeated.

#### 2.2.2. Corrosion Resistance Test

The UHPC was removed after 28 days of curing, dried at 80 °C for 2 days in an oven, weighed, and then put into a dry–wet cycle test machine for sulfate resistance testing. The solution was 5% Na_2_SO_4_. To keep the solution concentration unchanged, the water was changed every 30 days. Soaking was carried out for 16 h, followed by drying for 4 h and cooling for 2 h. The above-mentioned dry–wet cycles lasted for 22 h. The cycle numbers for testing the sulfate resistance of UHPC were 30, 60, 90, 120, and 150 cycles. After the number of cycles was reached, the test block was removed and dried at 80 °C for 1 day, and the mechanical properties were tested. The test block was compared to a test block that had the same curing age but was not subjected to dry–wet cycles, and the performance decline rate was obtained.

After 28 days of curing, the UHPC was removed and placed into the salt spray test chamber; the temperature of the box was 35 °C, the temperature of the saturated barrel was 47 °C, and the solution was 5% NaCl. The specific parameters are listed in Table 3. The salt spray test times were 7 days, 30 days, 60 days, 120 days, and 180 days. After reaching the test time, the UHPC was removed to measure the change of the flexural and compressive strength.

#### 2.2.3. Electrochemical Impedance Spectroscopy (EIS) Test

All samples subjected to electrochemical testing were dried at 80 °C for 8 h after 7 days of standard curing. After the samples were cooled, stainless-steel iron sheets with a length of 50 mm, width of 40 mm, and thickness of 0.5 mm were fixed on both ends of the UHPC test block using a conductive graphite adhesive. After the stainless-steel iron piece and test block were completely fixed for 1 d, an alternating current (AC) impedance test was carried out using the CHI660 electrochemical workstation. The amplitude was 0.5 mA and the test frequency was 0.1 Hz–1 Mhz. According to Keddam [37], the entire system was considered as the circuit shown in Figure 4.

#### 2.2.4. Three-Dimensional Structure of UHPFRC Test

In this experiment, the steel fiber distribution in the UHPC was investigated using the Zeiss Xradia 510 versa high-resolution three-dimensional X-ray microscope. A total of 1012 two-dimensional images were captured for the test block with a size of 30 × 30 × 40 mm, and the resolution was 1000 × 1024. Owing to the different densities of steel fibers and the UHPC matrices, when X-rays pass through the sample, they will exhibit different grayscales. The whiter part in the CT image is the steel fibers [38]. The steel fibers were labeled in the obtained 1102 CT images using the Avizo 2020.1 software, and three-dimensional reconstruction was carried out to obtain the steel fiber distribution in the UHPC. Subsequently, the Avizo 2020.1 software was used to remove interconnected fibers and calculate the connection probability. The specific process is shown in Figure 5.

## 3. Results and Discussion

### 3.1. Mechanical Properties of Developed UHPC

Figure 6a,b shows the flexural strength and compressive strength of UHPC, respectively, after mixing with hollow stainless-steel fiber and curing for 28 days. As can be clearly seen in Figure 6a, as the hollow stainless-steel fiber content increased, the flexural strength of UHPC also increased and reached the maximum of 42.3 MPa at a content of 2.5%. According to a previous study by Boulekbache [39], the reason for this is that the fiber plays a bridging role in the concrete. As the fiber content increases, this bridging role becomes more obvious, and thus the bending strength of UPHC is improved. This also explains why the improving effect of 0.5-mm hollow stainless-steel fiber on the flexural strength of UHPC diminishes when the fiber content is high: when the content is high, the fibers may become intertwined and the working efficiency of the fibers is reduced, which affects the workability of UHPC and results in weak bonding between the fibers and the concrete matrix [40]. 

In addition to the influence of the mixing amount, Figure 4 also shows the influence of the length–diameter ratio of the steel fiber on the mechanical properties of UHPC. As the length–diameter ratio becomes greater, the effect of the steel fiber on the mechanical properties of UHPC becomes more obvious [41]. At 2.5%, the effect of the steel fiber with an aspect ratio of 75 and 100 on the UHPC was higher by 55.8% and 91.4%, respectively, compared with that of the steel fiber with an aspect ratio of 60.

The influence trend of the steel fiber on the compressive strength of UHPC shown in Figure 6b is similar to that of the flexural strength shown in Figure 6a. Therefore, the content and aspect ratio of the steel fiber have obvious effects on the mechanical properties of UHPC.

Figure 7 shows the effect of copper-coated fiber, solid steel fiber, and hollow stainless-steel fiber on the flexural and compressive properties of UHPC at a content of 2%. The results reveal that the effect of hollow stainless-steel fiber on the flexural strength of UHPC is better than that of solid steel fiber. For example, the flexural strength of UHPC with 2% content of 0.5-mm solid steel fiber is 16.6 MPa, while the flexural strength of UHPC with 0.5-mm hollow stainless-steel fiber is 24.1 MPa, that is, higher by 45.2% compared with solid fiber. As shown in Figure 7a, as the fiber’s length–diameter ratio increased, the bending strength of UHPC first increased and then decreased, and the bending strength of the UHPC mixed with steel fiber with a length–diameter ratio of 75 is the largest. At this time, the bending strength of the UHPC mixed with solid steel fiber and hollow stainless-steel fiber was 24.8 MPa and 31.8 MPa, respectively. These phenomena are attributed to the fiber type and distribution. It is not difficult to imagine that, under the same volume and diameter, as the length–diameter ratio increases, the number of steel fibers with the same quality decreases. Therefore, appropriately increasing the length–diameter ratio (60–75) of fibers can significantly increase the number of steel fibers incorporated into the UHPC and make the steel fibers more evenly distributed in the UHPC. Thus, the UHPC can achieve high bending strength. However, with the continuous increase of the length–diameter ratio (75–100), the fibers added to the UHPC will interconnect and become entangled, which does not only reduce the fluidity of concrete and increase the difficulty of operation and construction, but also reduces the mechanical properties of UHPC.

### 3.2. Corrosion Resistance of Developed UHPC

#### 3.2.1. Dry–Wet Cycle Test

Figure 8a,b shows the change of the flexural strength of UHPC with copper-coated fiber and hollow stainless-steel fiber, respectively, after dry–wet cycle testing. As can be seen from the figure, the bending strength of UHPC with copper-plated fiber or hollow stainless-steel fiber reached the maximum at the curing age of 60–90 d. The reason for this is that the hydration reaction inside the concrete continued [42]; therefore, the strength of UHPC increased for a certain period of time. Owing to the influence of the dry–wet cycles, after the strength reached the maximum, the strength of the comparison sample under standard curing did not change, while the flexural strength of the UHPC subjected to dry–wet cycles decreased, and the maximum strength of the UHPC subjected to dry–wet cycles was lower than that of the sample subjected to standard curing.

Figure 8a shows that, after 150 days of dry–wet cycles, the flexural strength of the UHPC mixed with copper-coated fiber was reduced by 16.5% compared with the maximum value, and by 28.1% compared with the sample subjected to standard curing with the same age. According to Figure 8b, the UHPC with hollow stainless-steel fibers decreased only by 7.2% compared with the highest flexural strength after 150 days of dry–wet cycles, and decreased only by 6% compared with the sample subjected to standard curing. The performance difference of copper-coated fiber and hollow stainless-steel fiber in UHPC after the dry–wet cycle test is shown in Figure 8c. As can be seen, the bending strength of the UHPC with two steel fiber types first decreased and then increased as the dry–wet cycles increased. The intensity difference between the two was as high as 65.7% before the start of the dry–wet cycle test, and the difference was reduced to 38% when the intensity was maximum at 90 days. Therefore, the UHPC with hollow stainless-steel fiber achieved better performance in shorter time. As the dry–wet cycle period (90–150 days) progressed, the gap between the copper-coated steel fiber and hollow stainless-steel fiber became increasingly larger.

Figure 9 shows the change of the UHPC surface morphology. As can be clearly seen, the surface rust of the UHPC to which copper-coated steel fiber was added became increasingly more severe as the dry–wet cycles progressed, while the UHPC with hollow stainless-steel fiber did not change significantly. The corrosion of steel fibers in UHPC does not only result in inferior performance, but also causes rust stains on the surface, which is extremely detrimental to the promotion and use of UHPC. According to Lee et al. [43], modified sulfur-coated aggregate can significantly enhance the durability of Portland cement concrete. Additionally, according to this study, using stainless-steel fibers instead of ordinary steel fibers can also improve durability.

#### 3.2.2. Salt Spray Test

Figure 10 shows the bending strength of the UHPC with copper-coated steel fiber and hollow stainless-steel fiber after the salt spray test. The figure shows that as the salt spray test time increased, the bending strength of UHPC with two types of steel fiber first increased and then decreased. After 60 days, the bending strength of the UHPC reached the maximum, similar to the results of the dry–wet cycle test. At the beginning of the experiment, the hydration reaction in the concrete was still ongoing, and the strength exhibited an upward trend. As the experiment progressed, corrosive substances penetrated into the concrete and had a negative impact on the concrete matrix and steel fiber, resulting in the decrease of the UHPC strength.

Figure 11 shows improvement of bending strength of hollow stainless-steel-fiber UHPC compared with copper-coated-fiber UHPC after different times of salt spraying. Unlike the dry–wet cycles, the difference in the flexural strength of the UHPC with two types of steel fiber gradually increased with time. For example, the flexural strength of the UHPC to which hollow stainless-steel fiber was added was higher by 18.4% compared with that of solid steel fiber at 7 days of salt spraying, and the difference increased to 34% at the 180th day of the experiment. Therefore, the salt spray resistance of UHPC with hollow stainless-steel fiber is better than that of UHPC with copper-coated steel fiber. Additionally, Figure 12 shows the change of the surface morphology of UHPC after the salt spray test. As can be seen, the surface rust of the UHPC to which copper-coated steel fiber was added became increasingly more severe with the passage of time, and was more obvious compared with the dry–wet cycle test. The UHPC to which hollow stainless-steel fiber was added did not exhibit obvious change.

Based on the analysis of the dry–wet cycle and salt spray test results, the durability of UHPC with hollow stainless-steel fiber is better than that of UHPC with copper-coated steel fiber. Therefore, in practical engineering applications, with consideration to durability, copper-coated steel fiber is not suitable for engineering with high corrosion resistance.

### 3.3. Electrochemical Experiment

The AC impedance of UHPC was investigated to verify the effect of hollow stainless-steel fiber on the fiber network structure of UHPC and investigate the electrochemical corrosion resistance of UHPC with hollow stainless-steel fiber. Figure 13 shows the Nyquist diagram of UHPC mixed with 2% by volume of hollow stainless-steel fiber and solid steel fiber. In the figure, the horizontal axis is the real part, the vertical axis is the imaginary part, and the arc part represents the resistance of UHPC, which is represented by charge transfer resistance and electric double-layer capacitance in parallel.

As shown in Figure 13, the matrix resistance of UHPC gradually decreased with the steel fiber diameter. When the length of the steel fibers is the same, diameter reduction indicates that more steel fiber has been added. Therefore, the steel fibers in the UHPC are more densely arranged, the probability of interconnection increases, and the UHPC can be considered as a conductor. The same results are shown in Figure 14. As the steel fiber content increased, the matrix resistance of the UHPC decreased.

As shown in Figure 13 and Figure 14, the matrix resistance of the UHPC to which hollow stainless-steel fiber was added is greater than that of the UHPC to which solid steel fiber was added, regardless of the steel fiber specification or different mixing amounts. The specific reason is related to the heat balance of the conductor material when it is energized. Let the heat generation *Q_1_* of the energized conductor be as follows:(1)$$Q_{1}=K_{f}I^{2}\rho_{0}\left(1+\alpha \theta \right)Lt/S$$

Here, *K_f_* is the AC additional coefficient generated by the skin effect and proximity effect. For 50 Hz AC, *K_f_* = 1.02; *I* is the current; ρ_0_ is the resistivity of the conductor material at 0 °C; *L* is the length of the conductor material; *S* is the cross-sectional area of the conductor material; *α* is the resistance temperature coefficient; *θ* is the surface temperature of the conductor material; and *t* is the energization time.

Let the heat dissipation *Q_2_* of the surface temperature of the conductor material entering the steady state after power on be as follows:(2)$$Q_{2}=K_{t}ML\tau t$$
where *K_t_* is the comprehensive heat dissipation coefficient; *M* is the circumference of the conductor section; *L* is the length of the conductor; *τ* is the difference between the surface temperature of the conductor material and the ambient temperature; and *t* is the power on time.

When the surface temperature of the conductor is stable after power on, *Q*_1_ = *Q*_2_, that is, the heat generation is equal to the heat dissipation, and the following relationship holds:(3)$$K_{f}I^{2}\rho_{0}\left(1+\alpha \theta \right)Lt/S=K_{t}ML\tau t$$

Hence, the rated current *I_n_* of the conductor material is expressed as follows:(4)$$I_{n}=\sqrt{\frac{K_{t}M\tau_{d}S}{K_{f}\rho_{0}\left(1+\alpha \theta \right)}}$$
where *τ_d_* represents the maximum allowable temperature increase.

As can be seen from the conductor cross-sectional area s in Formula (3), the solid steel fiber is larger than the hollow stainless-steel fiber. Therefore, under the condition of the same allowable temperature increase *τ_d_*, the carrying capacity of the solid steel fiber of the same material is larger than that of the hollow stainless-steel fiber. Consequently, when hollow stainless-steel fiber is used in UHPC, the UHPC will exhibit higher resistance.

Additionally, hollow stainless-steel fiber is more evenly distributed in UHPC and cannot easily form a fiber network. The specific results were verified by the X-CT test discussed below.

### 3.4. X-ray CT Test

To analyze the hollow stainless-steel fiber and solid steel fiber networks in UHPC in a more detailed manner, this study conducted X-ray CT tests on the UHPC. Figure 15a–c shows the three-dimensional structure of UHPC with steel fibers (diameters of 0.3 mm, 0.4 mm, and 0.5 mm, respectively) added at 2% by volume. The figure shows the steel fibers that are not interconnected in the UHPC. To investigate the connection probability of steel fibers in the UHPC more intuitively, the connection probability of various types of steel fiber in UHPC was calculated using the AVIZO 2020.1 software and the results are shown in Figure 16.

Based on the probability of steel fiber interconnection shown in Figure 16, it can be found that the interconnection rate of 0.3-mm hollow stainless-steel fibers in UHPC is 80.1%, while the interconnection rate of 0.4-mm and 0.5-mm steel fibers is only 46.7% and 26.1%, respectively. As can be seen, the connection probability in the UHPC increased with the change of the steel fiber diameter, because the number of added steel fibers increased as the diameter decreased under the condition of constant volume content. Because the distribution space of the steel fibers was limited, the interconnection probability increased [44]. Moreover, it can be seen that the interconnection probability of hollow stainless-steel fiber is lower than that of solid steel fiber, regardless of the steel fiber specification. In conclusion, the distribution of hollow structures in UHPC is more advantageous.

With a content of 3.3 and 3.4, the UHPC with hollow stainless-steel fibers achieved better electrochemical corrosion resistance because the conductivity of steel fibers in hollow structures is lower than that in solid structures when electricity is conducted. Additionally, the probability of interconnecting hollow stainless-steel fibers in UHPC is lower, which makes it difficult to form a connected fiber network. Therefore, UHPC doped with hollow stainless-steel fibers has higher resistance and better electrochemical corrosion resistance.

## 4. Conclusions

This study systematically investigated the effects of hollow structural steel fiber and solid steel fiber on the mechanical properties, durability, and electrochemical properties of UHPC. The following conclusions were drawn:(1)The improving effect of hollow stainless-steel fiber on the mechanical properties of UHPC is better than that of solid steel fiber because the distribution of hollow stainless-steel fibers in concrete is more uniform, and the stress is more uniform when UHPC is subjected to loading. For example, the bending strength of UHPC with a content of 2% by volume of 0.5-mm hollow stainless-steel fiber is higher by 45.2% compared with that of solid steel fiber. Moreover, owing to the type of steel fiber and its distribution in the UHPC, the length–diameter ratio of the steel fiber is within a suitable range (approximately 75), which has a more significant impact on the mechanical properties of the UHPC.(2)In the durability experiments, the degradation of UHPC performance was caused by the combined effect of the concrete matrix and steel fiber degradation. Using stainless steel as the steel fiber material can greatly improve the durability of UHPC while avoiding surface rust stains. After dry–wet cycle testing, the flexural strength of the copper-coated-fiber-reinforced UHPC decreased by 21.9%, while the flexural strength of the UHPC with hollow stainless-steel fibers only decreased by 5.6%. When the salt spray test had run for seven days, the difference in the flexural strength between the two was 18.4%, but when the test was completed (180 days), the difference increased to 34%.(3)The addition of steel fiber led to a reduction in the resistance of the UHPC matrix. As more steel fibers are added, this phenomenon becomes more obvious, and may lead to electrochemical corrosion. However, compared with solid steel fiber, hollow stainless-steel fiber plays a positive role in reducing the resistance of UHPC, mainly because the carrying capacity of the hollow structure during conduction is smaller than that of the solid structure, and the hollow stainless-steel fiber is more evenly distributed in the UHPC and cannot easily form a fiber network.

## Figures and Tables

**Figure 1 materials-16-03612-f001:**
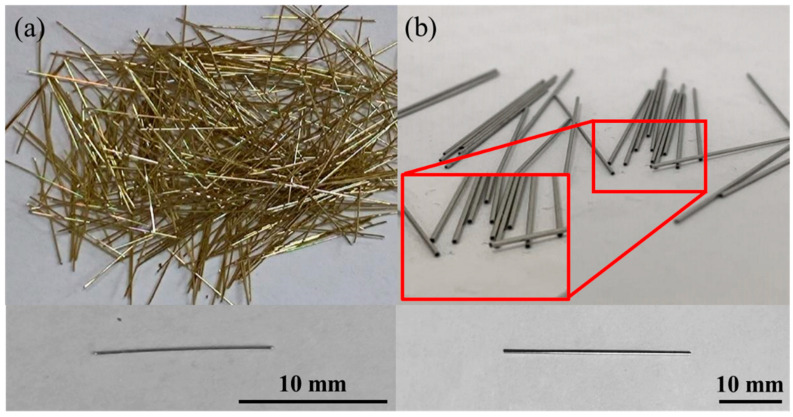
(**a**) Copper-plated steel fiber; (**b**) hollow stainless-steel fiber.

**Figure 2 materials-16-03612-f002:**
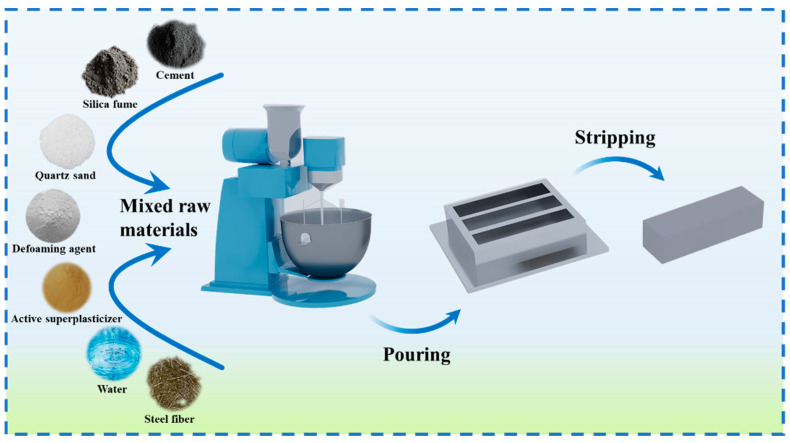
Preparation process of UHPC.

**Figure 3 materials-16-03612-f003:**
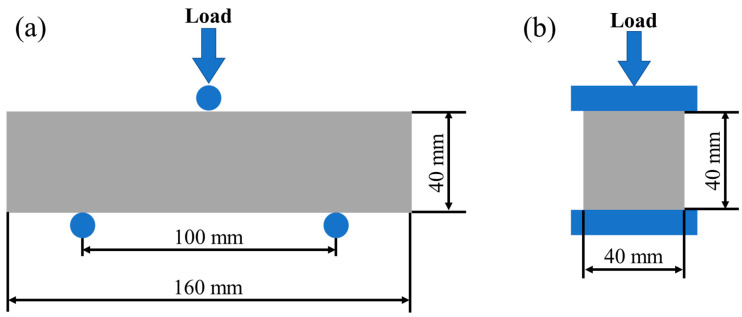
Schematic diagram of mechanical performance testing: (**a**) flexural strength; (**b**) compressive strength.

**Figure 4 materials-16-03612-f004:**
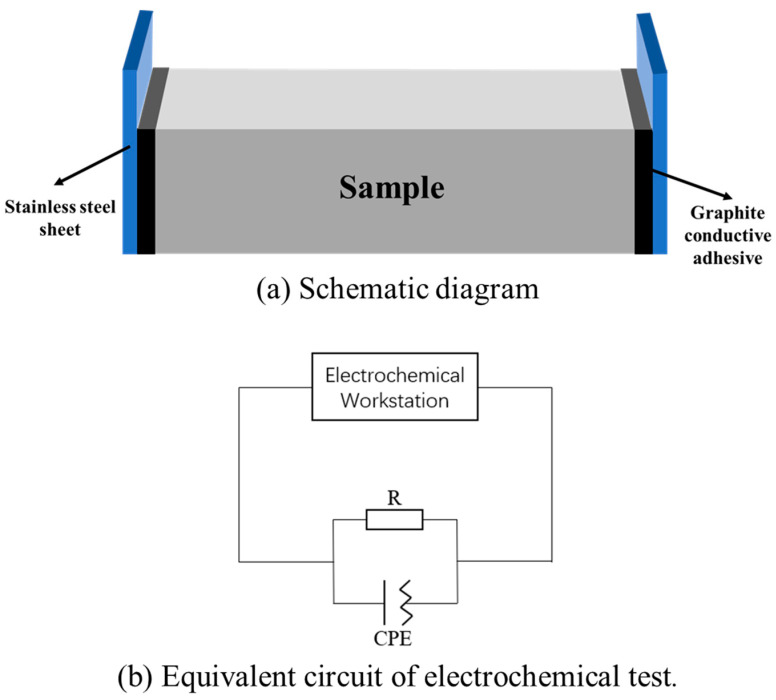
Procedure of electrochemical impedance measure.

**Figure 5 materials-16-03612-f005:**
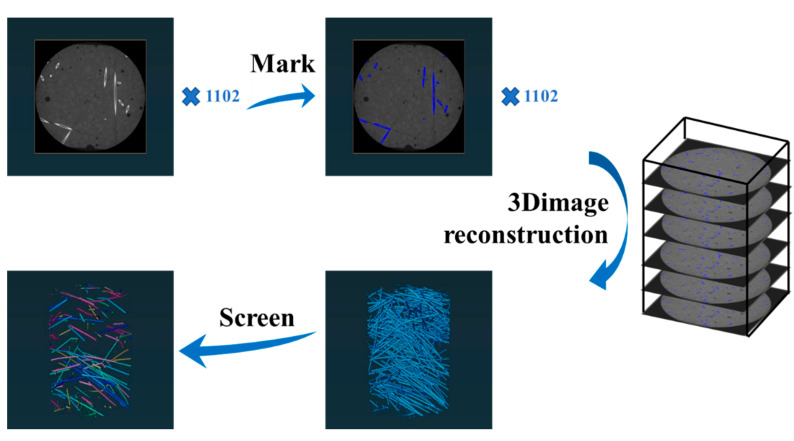
Process of rebuilding designed 3D UHPC structure.

**Figure 6 materials-16-03612-f006:**
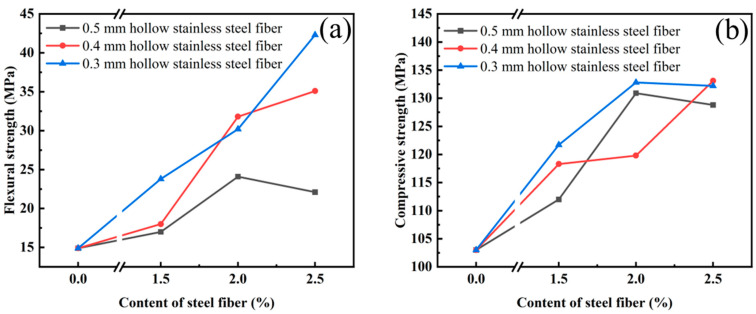
(**a**) Bending strength of UHPC with hollow stainless-steel fiber at 28 days; (**b**) compressive strength of UHPC with hollow stainless-steel fiber at 28 days.

**Figure 7 materials-16-03612-f007:**
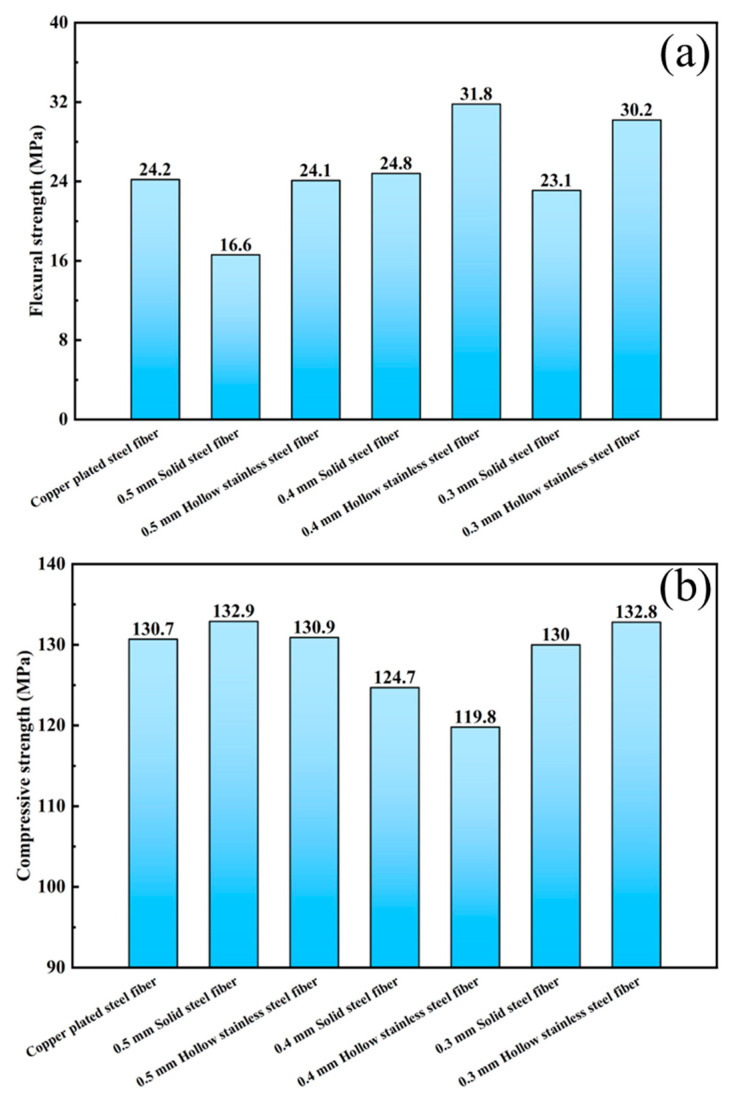
Effect of different fiber types on bending strength of UHPC at content of 2 vol%: (**a**) flexural strength; (**b**) compressive strength.

**Figure 8 materials-16-03612-f008:**
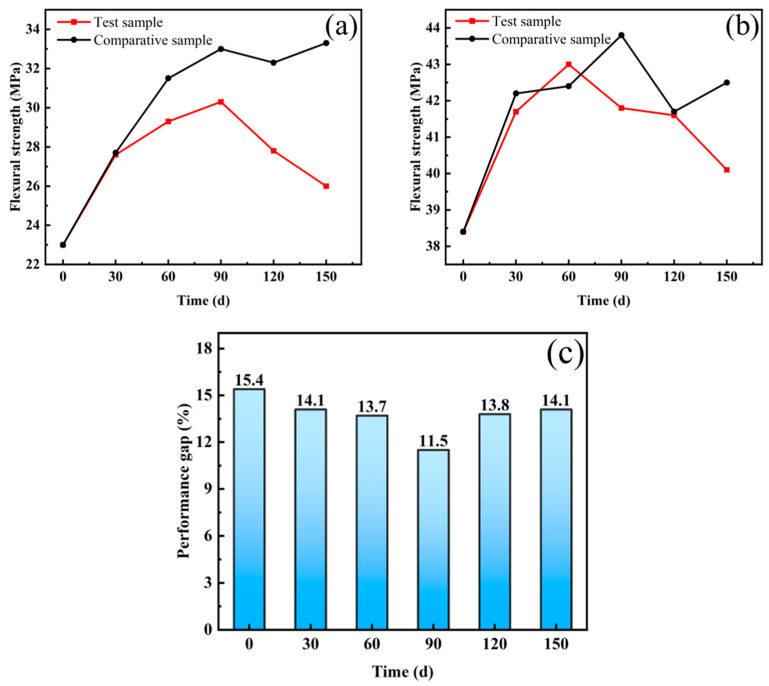
Dry–wet cycles: (**a**) flexural strength of UHPC mixed with copper-coated fiber after dry–wet cycles; (**b**) flexural strength of UHPC with hollow stainless-steel fiber after dry–wet cycles; (**c**) differences in mechanical properties of UHPC mixed with copper-plated steel fibers and hollow stainless-steel fibers after dry–wet cycling.

**Figure 9 materials-16-03612-f009:**
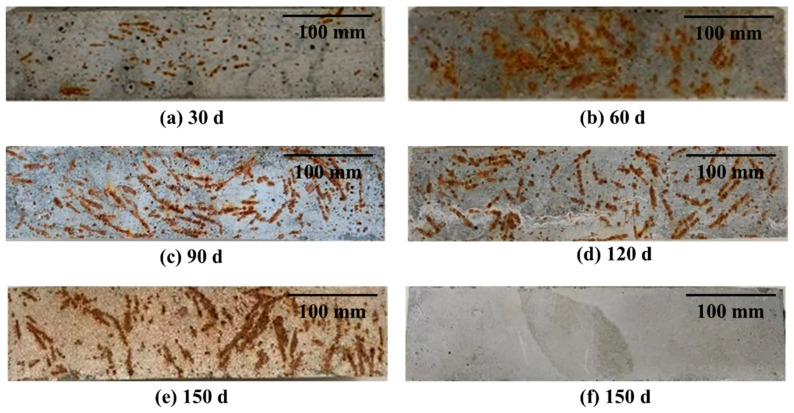
Surface morphology of UHPC after dry–wet cycles: (**a**) after 30 days with copper-plated steel fiber; (**b**) after 60 days with copper-plated steel fiber; (**c**) after 90 days with copper-plated steel fiber; (**d**) after 120 days with copper-plated steel fiber; (**e**) after 150 days with copper-plated steel fiber; (**f**) hollow stainless-steel fiber.

**Figure 10 materials-16-03612-f010:**
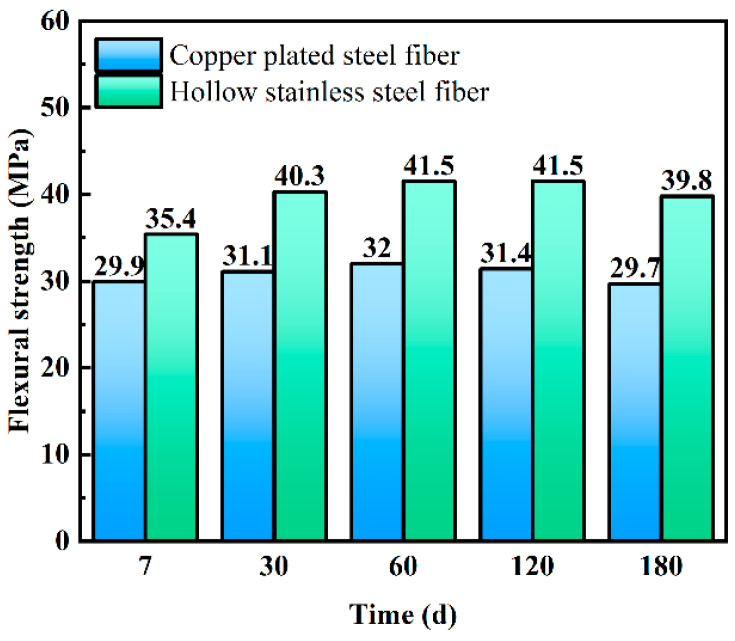
Effect of the salt spray test on bending strength of UHPC.

**Figure 11 materials-16-03612-f011:**
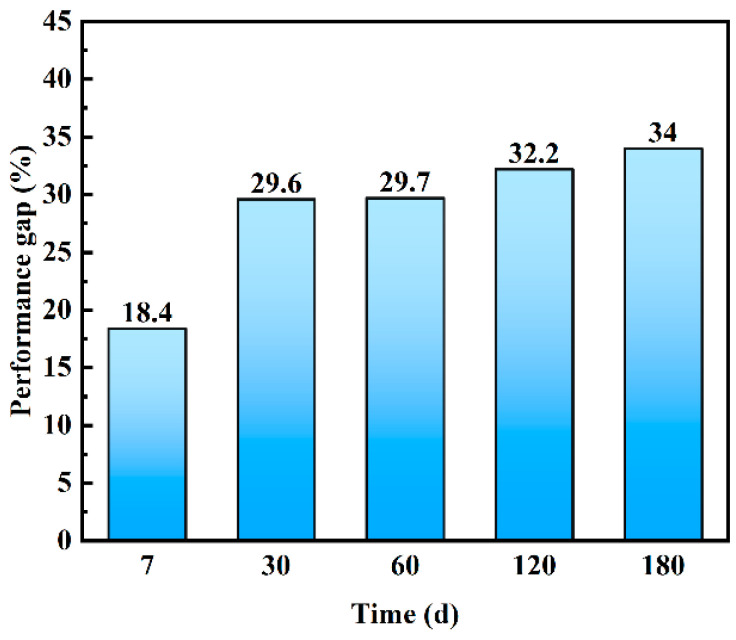
Differences between mechanical properties of UHPC in salt spray tests with copper-plated steel fibers and hollow stainless-steel fibers.

**Figure 12 materials-16-03612-f012:**
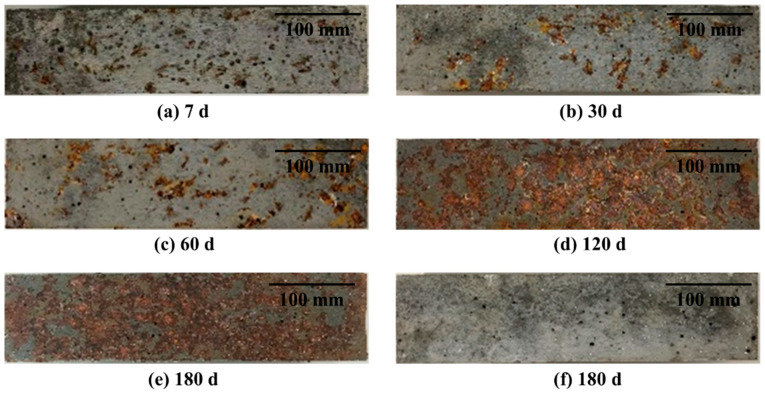
Surface morphology of UHPC after salt spray test: (**a**) copper-plated steel fiber at 7 days; (**b**) copper-plated steel fiber at 30 days; (**c**) copper-plated steel fiber at 60 days; (**d**) copper-plated steel fiber at 120 days; (**e**) copper-plated steel fiber at 180 days; (**f**) hollow stainless-steel fiber.

**Figure 13 materials-16-03612-f013:**
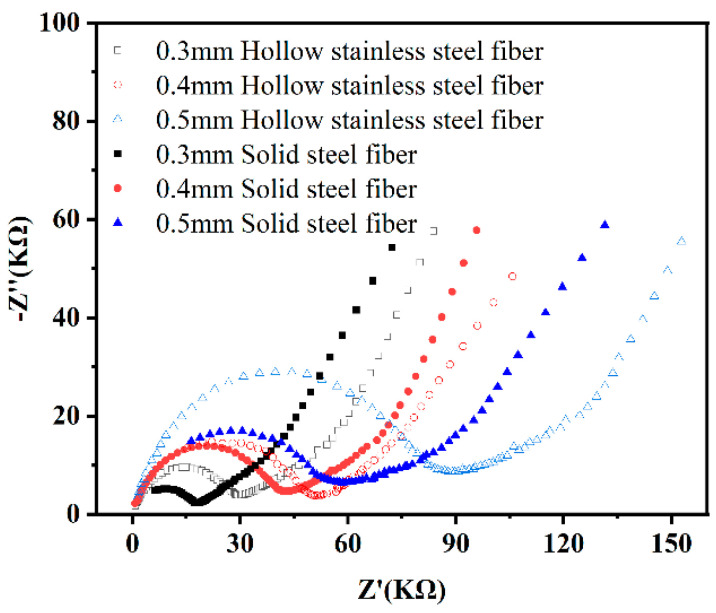
Nyquist diagram of UHPC with 2% solid and hollow stainless-steel fiber.

**Figure 14 materials-16-03612-f014:**
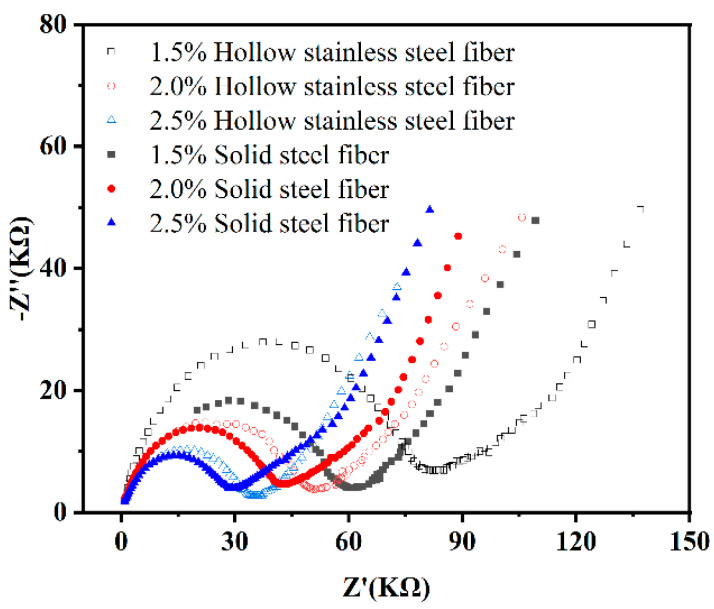
Nyquist diagram of UHPC with different solid steel fiber and hollow stainless-steel fiber content (0.4 mm).

**Figure 15 materials-16-03612-f015:**
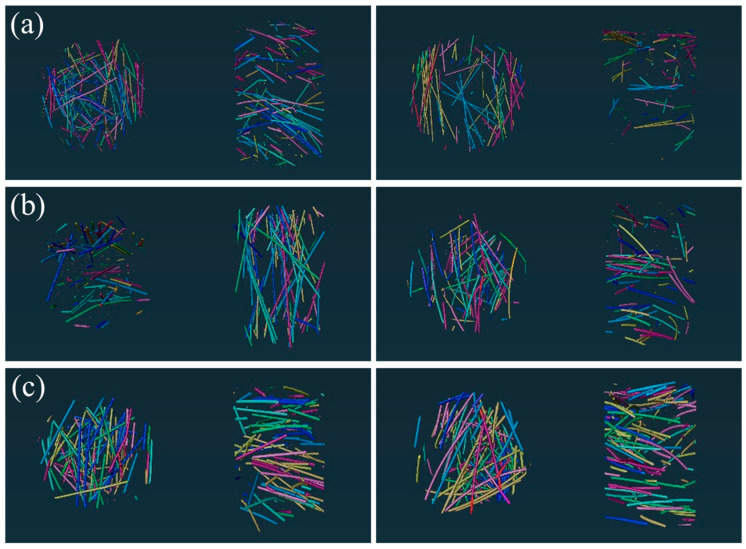
Three-dimensional structure of non-interconnected steel fibers in UHPC: (**a**) 0.3-mm hollow stainless-steel fiber (left) and solid steel fiber (right); (**b**) 0.4-mm hollow stainless-steel fiber (left) and solid steel fiber (right); (**c**) 0.5-mm hollow stainless-steel fiber (left) and solid steel fiber (right).

**Figure 16 materials-16-03612-f016:**
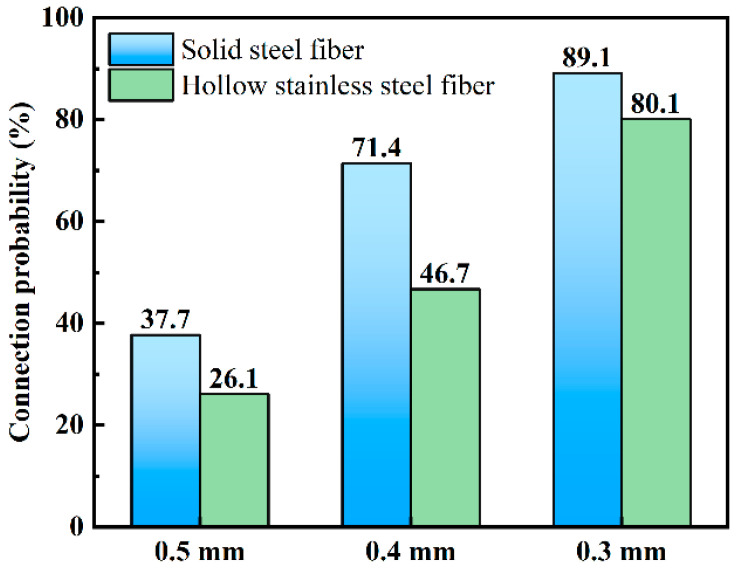
Connection probability of hollow stainless-steel fiber and solid steel fiber in UHPC.

**Table 1 materials-16-03612-t001:** Mix proportion of ultra-high performance fiber-reinforced concrete matrix (calculated by self-weight ratio).

Cement	Silica Fume	Quartz Sand (20-Grain)	Quartz Sand (40-Grain)	Quartz Sand (80-Grain)	Active Superplasticizer	Defoaming Agent	Water
1	0.25	0.375	0.375	0.375	0.007	0.00036	0.25

**Table 2 materials-16-03612-t002:** Specifications of various types of steel fiber.

Type	Diameter/Outer Diameter (mm)	Inside Diameter (mm)	Length (mm)	Length-Diameter Ratio	Tensile Strength (MPa)
Copper-plated steel fiber	0.2		13	65	>2800
Solid stainless-steel fiber	0.5		30	60	>1200
	0.4		30	75	>1200
	0.3		30	100	>1200
Hollow stainless-steel fiber	0.5	0.25	30	60	>1000
	0.4	0.20	30	75	>1000
	0.3	0.15	30	100	>1000

**Table 3 materials-16-03612-t003:** Relevant parameters of salt spray test chamber.

Saturation Barrel Temperature/°C	Salt Spray Deposition Rate	Spray Mode	Relative Humidity/%	Air Source/(kg/cm^2^)	Intake Pressure/MPa	Spray Pressure/MPa
35 ± 2	1–2 mL/80 cm^2^/h	Continuous spray	95 ± 5	8	0.3–0.4	0.05–0.17

## Data Availability

Data sharing is not applicable for this article.

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
