# Peer review of "Effect of Hollow 304 Stainless Steel Fiber on Corrosion Resistance and Mechanical Properties of Ultra-High Performance Concrete (UHPC)"

_materials, 2023, doi:10.3390/ma16103612_

Round 1

Reviewer 1 Report

Quite interesting article. Some comments are included in a file that you must take into account

Author Response

Response to Reviewer 1 Comments

Comments: It is a very interesting article, and very clear. Some indications are given to the authors on some points:

  1. The introduction should indicate the purpose of the research in practice, followed by the reason for the tests carried out.

Response: Thanks for the comment. We have changed the “introduction” in the revised manuscript. Added the purpose of the research in practice and the reasons for the tests.

  1. In materials, indicate the type of cement. Tests are carried out against sulfate attack cycles, and it is important to know if the cement was sulphoresistant (SR) and of what type

Response: Thanks for the comment. We have further introduced the types of cement in the revised manuscript: Portland cement (P·O425)

  1. Photograph 1 would give it in greater detail. If possible, with a very close photo to better see the copper plated fibers and the hollow Steel

Response: Thanks for the comment. The revised draft has added images of hollow steel fibers at high magnification. Add the following Figure 1 (corresponding to Figure 1 in the revised manuscript):

Figure 1. (a) Copper plated steel fiber; (b) Hollow stainless steel fiber

  1. In section 2.2.5 explain better and more understandably how the distribution of fibers is established on the basis of x-rays.

Response: Thanks for the comment. We have already revised the testing principles and processing methods of X-CT in the manuscript, as follows:

Due to the different densities of steel fibers and UHPC matrices, when X-rays pass through the sample, they will exhibit different grayscales. The whiter part in the CT image is the steel fibers. The 1102 CT images obtained were labeled with steel fibers using Avizo software and subjected to three-dimensional reconstruction to obtain the distribution of steel fibers in UHPC. Subsequently, the software was used to remove interconnected fibers and calculate the connection probability. The specific process is shown in Figure 2 (corresponding to Figure 5 in the revised manuscript).

Figure 2. The rebuilt process of the designed UHPC 3D structure

  1. In section 3.1, just before Figure 5, the interconnection rate is cited. I don't know if it's the site, and it's better with Figure 14. In addition, there seems to be a typo: 4.67% must be 46.7% according to Figure 14

Response: Thanks for the comment. The following modifications have been made in section 3.1 of the revised manuscript:

According to the interconnection probability of steel fibers shown in Figure 16 (originally Figure 14 has been transformed into Figure 16), the interconnection rate of 0.3mm hollow steel fibers in UHPC is 80.1%, while the interconnection rates of 0.4mm and 0.5mm steel fibers are only 46.7% and 26.1%, respectively.

  1. Section 3.3 is very specialized. I think we should put some more informative comment for readers not experts in this technique. In 3.4 at the end conclusions are drawn and that helps

Response: Thanks for the comment. We have added the following modifications after Section 3.4 in the revised draft:

Based on the above content, it can be seen that UHPC with hollow stainless steel fibers has better electrochemical corrosion resistance. This is because the conductivity of steel fibers in hollow structures is lower than that of solid structures when conducting electricity. In addition, the probability of interconnecting hollow stainless steel fibers in UHPC is lower, making it difficult to form a connected fiber network. Therefore, UHPC doped with hollow stainless steel fibers exhibits higher resistance and better electrochemical corrosion resistance. It's best to list the citations here.

Reviewer 2 Report

Please find an attached.

It is not bad.

Author Response

Response to Reviewer 2 Comments

Comments:This manuscript presents the results of investigating the effect of stainless steel fiber in ultrahigh performance concrete. Interesting and good. However, this reviewer thinks that the following changes should be made before publication is accepted.

  1. Introduction

- He et al. [10] in page 2 & Bei et al. [19] in page 3 : Is it correct?

- Authors need a detailed reason for choosing a hollow 304 stainless steel fiber.

- What is the hollow 304 stainless steel fiber? Key specifications need clarification.

Response: Thanks for the comment.

   1. For He et al. [10] in page 2, it was modified to Jiang et al. [10] found that with the increase of copper plated steel fiber content from 1% to 6%, the porosity of UHPC gradually decreased from 18.4% to 8.3%.

For Bei et al. [19] in page 3, it was modified to Bei et al. [19] found that After the cryogenic attack, the flexural strength of UHPC increased by 70.06% with the increase of length-to-diameter ratio of steel fiber, and the peak deflection increased from 0.501 mm 0.919 mm to 0.609 mm 1.302 mm.

    2. The reason for choosing hollow 304 stainless steel fiber is: under the same volume dosage, compared to solid steel fiber, cavitation in steel fiber can not only improve the mechanical properties of UHPC, but also reduce the use of steel fiber and reduce costs. At the same time, it can also bring about a weight reduction effect of about 1.5%, which is of great significance for the engineering application of UHPC. Moreover, cavitation in steel fibers can improve the distribution of steel fibers in UHPC and reduce the phenomenon of steel fiber agglomeration. Using stainless steel as a material for making steel fibers can greatly improve the durability of UHPC, while avoiding the yellowing phenomenon on the surface of UHPC caused by steel fiber corrosion. So this paper uses hollow stainless steel fibers.

   3. Hollow stainless steel fiber refers to a type of steel fiber made of 304 stainless steel. Compared with ordinary steel fiber, it has a hollow structure. The inner diameter of hollow stainless steel fibers is half of its outer diameter. This article uses three types of hollow stainless steel fibers with outer diameters, namely 0.5 mm, 0.4 mm, and 0.3 mm. The specific specifications can be seen in Table 1 (corresponding to Table 2 in the revised manuscript):

Table 1. Specifications of various types of steel fiber

Type

Diameter/Outer diameter (mm)

Inside diameter (mm)

Length (mm)

Length-diameter ratio

Tensile strength (MPa)

Copper plated steel fiber

0.2

13

65

> 2800

Solid stainless steel fiber

0.5

30

60

> 1200

0.4

30

75

> 1200

0.3

30

100

> 1200

Hollow stainless steel fiber

0.5

0.25

30

60

> 1000

0.4

0.20

30

75

> 1000

0.3

0.15

30

100

> 1000

  1. Table 1

- What do a 20 mesh, 40 mesh, and 80 mesh mean?

Response: Thanks for the comment. 20-mesh, 40-mesh, and 80-mesh, it indicates the particle size of quartz sand, which has been changed to 20 grain, 40 grain, and 80 grain in the revised version

  1. Table 2

- Is the length-diameter ratio of copper plated steel fiber correct?

Response: Thanks for the comment. The aspect ratio of copper plated steel fibers is 65, which has been corrected in Table 2 of the revised manuscript.

  1. Sample size & Test method

- Are all test samples 40 x 40 x 160mm in size?

- What is ‘national standard GB/T 39147-2020? International or Chinese standard?

- Overall, there are only result figures without experimental pictures. Add figures of test

process and results.

Response: Thanks for the comment.

   1. All the tested samples have dimensions of 40 x 40 x 160mm, but in the compressive strength test, the stress size of the samples is 40 x 40 x 40mm, as specified in Figure 3 of the revised draft.

   2. GB/T 39147-2020 is a Chinese standard. According to this standard, each data in this article is obtained by taking the average of the test results of three samples. When the data fluctuated over 15%, the data were rounded off. Moreover, when there were two data fluctuations exceeding 15%, the group of experiments was repeated.

   3. In the revised manuscript, specific UHPC production flowchart and testing diagram for flexural and compressive strength have been added, as shown in the following Figure 1, 2 (corresponding to Figure 2, 3 in the revised manuscript):

Figure 1. Preparation process of UHPC

Figure 2. Schematic diagram of mechanical performance testing: (a) Flexural strength; (b) Compressive strength

  1. Section 2.2.5

- Is sample 30 x 30 x 40mm in size?

- In Figure 3, why does it look like a circular section?

- It is necessary to explain what the figure shows.

Response: Thanks for the comment.

   1. Because during X-ray CT testing, in order to ensure that X-rays can penetrate the sample and obtain a clear image, the size of the sample is not easily too thick. So the sample size in section 2.2.5 is 30 x 30 x 40mm.

   2. Although the size of the test sample is 30 x 30 x 40mm, when conducting X-CT testing, the sample needs to rotate 360 ° along the central axis, resulting in a circular cross-section of the X-CT image.

   3. Further explanations have been provided in the revised manuscript, as follows:

   Owing to the different densities of steel fibers and the UHPC matrices, when X-rays pass through the sample, they will exhibit different grayscales. The whiter part in the CT image is the steel fibers. The steel fibers were labeled in the obtained 1102 CT images using the Avizo software, and three-dimensional reconstruction was carried out to obtain the steel fiber distribution in the UHPC. Subsequently, the Avizo software was used to remove interconnected fibers and calculate the connection probability. The specific process is shown in Figure 3.(corresponding to Figure 5 in the revised manuscript).

Figure 3. The rebuilt process of the designed UHPC 3D structure

  1. Figure 4 & Figure 8

- Mpa -> MPa

- hollow steel fiber -> hollow stainless steel fiber : Correct the text and figures in the entire manuscript so that the readers are not confused.

- Show the test process and results.

- How many samples per experiment?

Response: Thanks for the comment.

   1. The issues of —Mpa —>Mpa, —hollow steel fiber —>hollow stainless steel fiber have been revised in the revised draft.

   2. The testing process and results are shown in Figure 4 (corresponding to Figure 3 in the revised manuscript), and have been provided for annotation in the revised draft:

Figure 4. Schematic diagram of mechanical performance testing: (a) Flexural strength; (b) Compressive strength

   3. In this article, three samples were used in each test. The flexural strength and compressive strength of the UHPC were measured according to the Chinese national standard GB/T 39147-2020. Data were obtained by averaging the test results of the three samples. When the data fluctuated over 15%, the data were rounded off. Moreover, when there were two data fluctuations exceeding 15%, the group of experiments was repeated.

  1. Figure 5

- Edit the name of last bar graph. (0.4mm -> 0.3mm)

Response: Thanks for the comment. Corrected in the revised draft.

  1. Figure 6

- Title of y-axis is required.

- Please clearly distinguish between ‘Test sample’ and ‘Comparative sample’.

- Figure 6(c) needs some more specific explanation.

Response: Thanks for the comment. Figure 5 (original Figure 6) has been revised in the revised draft (corresponding to Figure 8 in the revised manuscript).

Figure 5. Dry-wet cycles: (a) flexural strength of UHPC mixed with copper-coated fiber after dry-wet cycles; (b) flexural strength of UHPC with hollow stainless steel fiber after dry-wet cycles; (c) differences in mechanical properties of UHPC mixed with copper plated steel fibers and hollow stainless steel fibers after dry wet cycling

  1. Figure 7

- It would be nice to express the scale so that readers can clearly check the surface condition.

Please refer to Figure 6 in the following paper: Lee, S.-H., Hong, K.-N., Park, J.-K. and Ko, J. (2014), “Influence of Aggregate Coated with Modified Sulfur on the Properties of Cement Concrete”, Materials, 7, 4739-4754.

Response: Thanks for the comment. We have referred to the above citations and applied them in the revised manuscript 3.2.1.

Figure 6. Surface morphology of UHPC after dry-wet cycles: (a) after 30 days with copper-plated steel fiber; (b) after 60 days with copper-plated steel fiber; (c) after 90 days with copper plated steel fiber; (d) after 120 days with copper plated steel fiber; (e) after 150 days with copper plated steel fiber; (f) hollow stainless steel fiber.

  1. Conclusions & Abstract

- Conclusions and abstract only list experimental results. It is requested to propose a practical mix proportion suitable for the purpose

Response: Thanks for the comment. The abstract and conclusion in the revised manuscript have been further improved.

Reviewer 3 Report

Effect of hollow stainless steel fiber on corrosion resistance and mechanical properties of ultra high performance concrete (UHPC)

Article is interesting. Few observations are given below;

Abstract need revision with some quantitative results.

Some more latest studies are required in the introduction section to further highlight the importance of this study. Please consider following to include.

Zhong, H., Chen, M., & Zhang, M. (2023). Effect of hybrid industrial and recycled steel fibres on static and dynamic mechanical properties of ultra-high performance concrete. Construction and Building Materials, 370, 130691.

Khan, K., Ahmad, W., Amin, M. N., & Nazar, S. (2022). Nano-silica-modified concrete: A bibliographic analysis and comprehensive review of material properties. Nanomaterials, 12(12), 1989.

Table 1, how mix proportions were designed?

Figure 1. Please provide subcaptions.

Section 2.2.1 which standards were followed?

Its better to inlcude prepartion of specimens with photographs.

The qualitation of graphs in results section is not good.

Authors must summarized results in more systematic way with reference to the previous studies.

Also, Conclusions are too limited to proof the significant outcome of this study.

There is need to further improve the english grammer. 

Author Response

Response to Reviewer 3 Comments

Comments: Effect of hollow stainless steel fiber on corrosion resistance and mechanical properties of ultra high performance concrete (UHPC). Article is interesting. Few observations are given below;

  1. Abstract need revision with some quantitative results.

Response: Thanks for the comment. The abstract in the revised manuscript have been further improved.

  1. Some more latest studies are required in the introduction section to further highlight the importance of this study. Please consider following to include.

Zhong, H., Chen, M., & Zhang, M. (2023). Effect of hybrid industrial and recycled steel fibers on static and dynamic mechanical properties of ultra-high performance concrete. Construction and Building Materials, 370, 130691.

Khan, K., Ahmad, W., Amin, M. N., & Nazar, S. (2022). Nano-silica-modified concrete: A bibliographic analysis and comprehensive review of material properties. Nanomaterials, 12(12), 1989.

Response: Thanks for the comment. I have referred to the above citations and applied them in the revised Introduction.

  1. Table 1, how mix proportions were designed?

Response: Thanks for the comment. Cement, as the material with the largest amount of sample preparation in this article, is set to 1. The usage of other materials can be obtained by comparing them with cement. The specific usage of materials is as follows Table 1:

Table 1. Materials used for ultra-high performance fiber reinforced concrete matrix

cement

Silica fume

Quartz sand (20-grain)

Quartz sand (40-grain)

Quartz sand (80-grain)

Active

Superplasticizer

Defoaming agent

water

880 g

220 g

330 g

330 g

330 g

6.4 g

0.32 g

220 g

  1. Figure 1. Please provide subcaptions.

Response: Thanks for the comment. The modification has been made to Figure 1 (corresponding to Figure 1 in the revised manuscript).

Figure 1. (a) Copper plated steel fiber; (b) Hollow stainless steel fiber

  1. Section 2.2.1 which standards were followed?

Response: Thanks for the comment. Section 2.2.1 follows the Chinese standard GB/T 39147-2020. The flexural strength of the sample was tested using a three-point bending test with a span of 100 mm, and a loading rate of 50 N/s. In the compressive strength test of the UHPC, the loading rate was 2400 N/s, as shown in Figure 2. Three samples were used in each test. The flexural strength and compressive strength of the UHPC were measured according to the Chinese national standard GB/T 39147-2020. Data were obtained by averaging the test results of the three samples. When the data fluctuated over 15%, the data were rounded off. Moreover, when there were two data fluctuations exceeding 15%, the group of experiments was repeated.

Figure 2. Schematic diagram of mechanical performance testing: (a) Flexural strength; (b) Compressive strength

  1. Its better to inlcude prepartion of specimens with photographs.

Response: Thanks for the comment. The experimental flowchart has been added to the revised draft, as shown in the following Figure 3 (corresponding to Figure 2 in the revised manuscript):

Figure 3. Preparation process of UHPC

  1. The qualitation of graphs in results section is not good.

Response: Thanks for the comment. The quality of the image has been modified in the revised manuscript.

  1. Authors must summarized results in more systematic way with reference to the previous studies.

Response: Thanks for the comment. The results section has been modified.

  1. Also, Conclusions are too limited to proof the significant outcome of this study.

Response: Thanks for the comment. The conclusion section has been modified, as follow:

This study systematically investigated the effects of hollow structural steel fiber and solid steel fiber on the mechanical properties, durability, and electrochemical properties of UHPC. The following conclusions were drawn:

  • The improving effect of hollow stainless steel fiber on the mechanical properties of UHPC is better than that of solid steel fiber, because the distribution of hollow stainless steel fibers in concrete is more uniform, and the stress is more uniform when UHPC is subjected to loading. For example, the bending strength of UHPC with a content of 2% by volume of 0.5-mm hollow stainless steel fiber is higher by 45.2% compared with that of solid steel fiber. Moreover, owing to the type of steel fiber and its distribution in the UHPC, the length–diameter ratio of the steel fiber is within a suitable range (approximately 75), which has a more significant impact on the mechanical properties of the UHPC.
  • In the durability experiments, the degradation of UHPC performance was caused by the combined effect of the concrete matrix and steel fiber degradation. Using stainless steel as the steel fiber material can greatly improve the durability of UHPC while avoiding surface rust stains. After dry–wet cycle testing, the flexural strength of the copper-coated-fiber-reinforced UHPC decreased by 21.9%, while the flexural strength of the UHPC with hollow stainless steel fibers only decreased by 5.6%. When the salt spray test had ran for seven days, the difference in the flexural strength between the two was 18.4%, but when the test was completed (180 days), the difference increased to 34%.
  • The addition of steel fiber led to a reduction in the resistance of the UHPC matrix. As more steel fibers are added, this phenomenon becomes more obvious, and may lead to electrochemical corrosion. However, compared with solid steel fiber, hollow stainless steel fiber plays a positive role in reducing the resistance of UHPC, mainly because the carrying capacity of the hollow structure during conduction is smaller than that of the solid structure, and the hollow stainless steel fiber is more evenly distributed in the UHPC and cannot easily form a fiber network.

Round 2

Reviewer 2 Report

I think it's well corrected and improved.

I think it's well corrected and improved.